# Strainer-Separable TiO_2_ on Halloysite Nanocomposite-Embedded Alginate Capsules with Enhanced Photocatalytic Activity for Degradation of Organic Dyes

**DOI:** 10.3390/nano12142361

**Published:** 2022-07-10

**Authors:** Jewon Lee, Sicheon Seong, Soyeong Jin, Jaeyong Kim, Youngdo Jeong, Jaegeun Noh

**Affiliations:** 1Department of Convergence of Nanoscience, Hanyang University, 222 Wangsimni-ro, Seongdong-gu, Seoul 04763, Korea; chu0254@naver.com; 2Department of Chemistry, Hanyang University, 222 Wangsimni-ro, Seongdong-gu, Seoul 04763, Korea; ssc09122@hanyang.ac.kr (S.S.); truejin@hanyang.ac.kr (S.J.); 3Center for Biomaterials, Biomedical Research Institute, Korea Institute of Science and Technology (KIST), Seoul 02792, Korea; 4Department of Physics, Hanyang University, 222 Wangsimni-ro, Seongdong-gu, Seoul 04763, Korea; kimjy@hanyang.ac.kr; 5Department of HY-KIST Bio-Convergence, Hanyang University, 222 Wangsimni-ro, Seongdong-gu, Seoul 04763, Korea; 6Institute of Nano Science and Technology, Hanyang University, 222 Wangsimni-ro, Seongdong-gu, Seoul 04763, Korea

**Keywords:** photocatalyst, halloysite nanotubes, titanium oxide, alginate capsule, strainer separation, photocatalytic effect, degradation, organic dyes

## Abstract

Photocatalysis driven by natural sunlight is an attractive approach to removing pollutants from wastewater. Although TiO_2_–based photocatalysts using various support nano-materials with high catalytic activity and reusability have been developed for purifying wastewater, the centrifugal separation methods used for the nanocatalysts limit their use for treating large amounts of water. Here, we prepared a TiO_2_ nano-catalyst supported on a halloysite nanotube (HNT)-encapsulated alginate capsule (TiO_2_@HNT/Alcap) to recapture the catalysts rapidly without centrifugation. The structure of TiO_2_@HNT/Alcap was characterized by X-ray diffraction, SEM, and TGA. In our system, the combination of HNTs and alginate capsules (Alcaps) improved the efficiency of adsorption of organic pollutants to TiO_2_, and their milli = meter scale structure allowed ultra-fast filtering using a strainer. The TiO_2_@HNT/Alcaps showed ~1.7 times higher adsorption of rhodamine B compared to empty alginate capsules and also showed ~10 and ~6 times higher degradation rate compared to the HNT/Alcaps and TiO_2_/Alcaps, respectively.

## 1. Introduction

Photocatalysis driven by inexhaustible solar energy may be well-suited to the issue of wastewater purification in the textile industry through an eco-friendly and efficient method [1,2]. As a photocatalyst, anatase TiO_2_ nanoparticles have shown high activity for the degradation of organic dyes and toxic molecules in the aquatic environment [3,4,5]. However, their innate toxicity incurs other burdens for environmental health and safety and limits their large-scale use. Thus, to minimize environmental burden and optimize function, both the TiO_2_ catalyst retrieval strategy and catalytic activity need improvement for practical use. 

For enhancing TiO_2_ catalytic activity, catalyst immobilization on Appendix A has been applied to prevent aggregation and sustain activity. Among the many Appendix A available, halloysite nanotubes (HNTs) that consist of aluminosilicate clay minerals with a tubular bilayer structure have contributed to the enhanced catalytic activity of TiO_2_ [6,7,8,9]. Additionally, it is spotlighted as a naturally occurring porous material for activity enhancement through the adsorption of contaminants induced by electric charges on the surface and inside of HNTs. The negatively charged Si-O-Si surface structure and positively charged Al-OH internal structure can adsorb cationic dyes (methylene blue, neutral red, crystal violet, and malachite green) and anionic dyes (congo red and methyl orange), respectively. [10,11,12,13,14,15,16] Although several TiO_2_@HNTs composites have shown enhanced activity in photocatalytic degradation of organic pollutants [17,18,19,20,21,22,23], their nanosize still requires complicated separation processes, hindering the catalysts’ retrieval.

From the perspective of reuse and easy separation, alginate has been employed as a Appendix A. Alginate, a component extracted from natural algae, is a linear copolymer composed of (1,4) linked α-L-guluronic acid and β-D-mannuronic acid [24]. In previous research, alginate has adsorbed dyes and heavy metals, purifying the wastewater [25,26]. For usability, gelated alginate particles have been prepared by combining alginate polymers via cations [27]. By adding the catalysts during the gel formation, TiO_2_ nanoparticles can be encapsulated on the gelated particle. These alginate capsules containing TiO_2_ improved the degradation efficiency of methylene blue and methyl orange under UV conditions and could be easily separated by strainer [28,29] in a short time. Despite the easy separation resulting from the regulated millimeter size of alginate gel, their low activity requires the use of a large amount of catalyst, decreasing the purifying efficiency. For practical use of the photocatalyst, both the enhancement of catalytic activity and the feature of high reusability are required.

In this study, we prepared an alginate capsule including TiO_2_ nanoparticles on HNTs (TiO_2_@HNTs/Alcaps) to achieve both improved photocatalytic activity and ultra-fast catalyst separation (Figure 1). There are different chemical methods used for the synthesis of uniform micro- and nano-particles and their composites: for example, the micro-emulsion method [30], co-precipitation method [31], sol-gel method [32], and sol-vothermal/hydrothermal method with different surfactant and caping agents [33]. In this work, a TiO_2_ on HNTs (TiO_2_@HNTs) composite was synthesized by a simple sol-gel method using titanium(IV) isopropoxide (TTIP). The mixed solution of alginate and TiO_2_@HNTs was added dropwise to the CaCl_2_ solution, forming millimeter-sized capsules by crosslinking. Our capsules can be rapidly separated from the reaction mixture by a strainer with a sub-millimeter mesh size. Since the centrifugation of a large amount of wastewater would require huge equipment and energy in quantity, the nano-catalysts needed for separation processes including centrifugation offer limited use in large-scale processing. However, our catalysts of millimeter size can be easily separated from the wastewater without centrifugation, so that they can be applied for practical use regardless of the scale of operation. Additionally, comparing to the other catalysts of millimeter size, our catalysts showed much higher photocatalytic activity, allowing for the use of smaller quantities of catalyst. The dye adsorption efficiency improved by 1.7 times through the encapsulation of TiO_2_@HNTs compared to the alginate gel. The combination of TiO_2_ nanoparticles, HNTs, and encapsulation in alginate gel promoted photocatalytic activity up to ~10 and ~6 times compared to HNT/Alcaps and TiO_2_/Alcaps, respectively, due to their synergistic effect.

## 2. Materials and Methods

### 2.1. Preparation of TiO_2_@HNTs Composite

TiO_2_ nanoparticles on the HNT surface were synthesized by the sol-gel method [19,34] (Figure 2a). In brief, 5 mL of TTIP was dissolved in 24 mL of isopropanol. To the 16 mL of mixture, 16 mL of water was added. Then, 16 mL of isopropanol and 0.2 mL of nitric acid were added dropwise to the round-bottom flask containing the reaction mixture, with vigorous stirring. After 2 h stirring, 1.3 g of HNT was added, with 2 h stirring. Through 1-day aging at room temperature, the TiO_2_@HNTs were formed. Through centrifugation (8000 rpm for 10 min) and several washing steps with ethanol/water mixture (1:1), the TiO_2_@HNTs were purified. To gain the powder form of catalysts, the samples were dried in a vacuum oven at 85 °C for 12 h and then were ground and calcined at 350 °C for 2 h (2 °C/min rates) under ambient conditions. The synthesized catalysts were analyzed by high-resolution TEM (Appendix A) and showed successful immobilization and the characteristic anatase structure of TiO_2_. From the XRD spectrum of TiO_2_ nanoparticles, HNT, and TiO_2_@HNT, we could also confirm the adsorption of TiO_2_ nanoparticles on the HNTs (Appendix A). In the spectrum of TiO_2_@HNT, the typical peaks of HNT and TiO_2_ crystal structure were observed.

### 2.2. Preparation of TiO_2_@HNTs/Alcaps Composite

TiO_2_@HNTs were readily immobilized in calcium alginate capsules (~2.5 ± 0.2 mm) using a simple dropping technique with a syringe [28,35] (Figure 2b). A 2.0 g portion of the TiO_2_@HNTs was dispersed in DI-water (~100 mL) to attain a 2% solution (*w*/*w*). The solution was heated to 60 °C, with stirring for 30 min. A 2 g portion of sodium alginate was dissolved in the solution to result in the mass ratio between alginate and TiO_2_@HNTs being 1:1. Using a syringe, we dropped the polymer solution into a 0.1 M CaCl_2_ solution during gentle stirring at room temperature. After aging for several hours, the obtained capsules were washed with DI-water and dried in an oven at 60 °C for 12 h.

### 2.3. Structural Analysis of TiO_2_@HNT/Alcaps 

The morphology of capsules was analyzed using a scanning field emission electron microscope (FE-SEM, Hitachi, s-4800 instrument). For the measurement, the TiO_2_@HNT/Alcap sample was freeze-dried and then coated with platinum. To examine thermal degradation characteristics of alginate gel and TiO_2_@HNTs/Alcaps, thermogravimetric analysis (TGA) was performed using the SDT Q600, TA Instruments (Heating: 10 °C/min, N_2_ = 100 mL/min, 800 °C). UV-vis measurements were performed with Evolution 60S (Thermo Fisher Scientific: Waltham, MA, USA).

### 2.4. Measurement of Rhodamine B Adsorption 

The dye adsorption experiments were evaluated by adding 0.1 g of alginate, HNTs/Alcap, TiO_2_/Alcaps, and TiO_2_@HNTs/Alcaps to 20 mL of 5.0 mg/L rhodamine B solution. The absorbance at 554 nm of the rhodamine B solution was monitored under stirring (800 rpm) conditions for 1 h. To calculate the removal efficiency (%), q_t_ (the rhodamine B adsorption capacity, unit: mg/g), q_e_ (the rhodamine B adsorption capacity at equilibrium, unit: mg/g), and *k*_2_ (rate constant, unit: g/mg min), we used the following the equations [36].
(1)$$\mathrm{Removal}\;\mathrm{efficiency}\;(\%)=\frac{C_{0}-C_{e}}{C_{0}}\;\times \;100\;$$
(2)$$q_{e}\;(\mathrm{mg}/g)=\frac{\left(C_{0}-C_{e}\right)V}{m}$$
(3)$$q_{t}\;(\mathrm{mg}/g)=\frac{\left(C_{0}-C_{t}\right)V}{m}$$
(4)$$\frac{t}{q_{t}}=\frac{1}{k_{2}q_{e}{}^{2}}+\frac{1}{q_{e}}t$$
where C_0_ and C_t_ are the concentrations of dye at initial and t time (unit: mg/L), respectively; m and V are the weight of catalyst (unit: g) and the volume of dye solution (unit: L), respectively.

### 2.5. Photodegradation of Rhodamine B Using the Capsules

For the photocatalytic dye degradation of rhodamine B, 0.1 g of alginate, HNTs/Alcaps, TiO_2_/Alcaps, and TiO_2_@HNT/Alcaps were added to 20 mL of 5.0 mg/L rhodamine B solution, respectively. The solutions were stirred in the dark for 1 h. Then, we applied UV light (250 W, 356 nm) using a lamp (Ushio-SP9) to monitor the change of absorbance induced by the dye degradation. The dye degradation efficiency (%) and the rate constant were obtained using the equations below [37].
(5)$$\mathrm{Degradation}\;\mathrm{efficiency}\;(\%)=\frac{C_{0}-C}{C_{0}}\;\times \;100\;$$
(6)$$\ln\frac{C_{0}}{C}=k_{1}t$$
where C_0_ and C are the dye concentrations at initial and final times (unit: mg/L) and *k*_1_ is the rate constant of the pseudo first order (unit: min^−1^). 

## 3. Results and Discussion

### 3.1. Morphology of TiO_2_@HNT/Alcaps Capsule

To confirm the encapsulation of TiO_2_@HNTs in the alginate gel, we observed their morphology using FE-SEM. In the images of TiO_2_@HNT/Alcaps, the smallest capsule size was ~2.5 ± 0.2 mm (Figure 3a). Although the TiO_2_@HNTs catalysts were barely seen on the capsule surface (Figure 3b,c), a large number of catalysts were observed in the inside of the capsule (Figure 3d–f). TiO_2_@HNTs with lengths of 200–500 nm were observed and TiO_2_ nanoparticles existed on the HNTs’ surface. We assumed that this morphology of TiO_2_@HNT/Alcaps could prevent the sweep of photocatalysts from the capsules. The TEM images of the TiO_2_@HNT clearly show the immobilization of TiO_2_ nanoparticles on the HNTs and the crystal structure of TiO_2_ nanoparticles (Appendix A). In the image, ~10 nm–sized TiO_2_ nanoparticles were immobilized on the surface of HNTs, and the crystal structure of the nanoparticles showed the (101) phase, typical anatase TiO_2_ structure.

### 3.2. Thermogravimetric Analysis of TiO_2_@HNT/Alcaps

The thermal degradation characteristics of TiO_2_@HNT/Alcaps and pure alginate gel as controls were measured through TGA analysis (Figure 4). In the range of 0 to 200 °C, the alginate gel and TiO_2_@HNT/Alcaps showed decompositions of 9.83% and 5.65%, respectively, indicating the evaporation of water adsorbed in the alginate capsule [38]. We observed decompositions of 34.59% of alginate gel and 17.20% of TiO_2_@HNT/Alcaps in the range of 200 to 300 °C. This decomposition results from the decarboxylation in the glycoside chain of alginate [39]. In the range of 450–550 °C and the resultant dihydroxylation, 5.86% of alginate gel and 5.07% of TiO_2_@HNT/Alcaps were decomposed [40,41]. We assumed that the reason for the higher decomposition of alginate gel is that the pure alginate gel possesses more alginate monomer including hydroxyl groups than TiO_2_@HNT/Alcaps at the same weight. The remaining amounts were 33.3% for alginate gel and 59.5% for TiO_2_@HNT/Alcaps. From the 26.2% difference between residues, we estimated the weight % of TiO_2_@HNTs in TiO_2_@HNT/Alcaps. Based on the peak intensity of the XRD spectrum of TiO_2_, HNTs, and TiO_2_@HNTs, we could roughly confirm that our TiO_2_@HNT/Alcaps consisted of 9% HNTs, 17% TiO_2_, and 74% alginate gel.

### 3.3. Adsorption and Kinetic Studies of Rhodamine B Using TiO_2_@HNT/Alcaps

After material characterizations, we probed the rhodamine B adsorption and decomposition kinetics of the pure alginate gel, HNTs/Alcap, TiO_2_/Alcap, and TiO_2_@HNT/Alcap under dark conditions to prevent the photodegradation of dyes (Figure 5a). In the adsorption curve of rhodamine B for each sample, the adsorption amount increased significantly up to 10 min and stabilized after 30 min. For all samples, the rate of rhodamine B removal rapidly increased initially but steadily decreased as it reached an equilibrium state. In the early stage, a large amount of rhodamine B was adsorbed promptly because of the sufficient availability of adsorption sites on the alginate gels and HNT surface. Table 1 summarizes the adsorption amount and rate constants. The adsorption curve of the dye mainly follows pseudo-second-order kinetics (Figure 5b) [42]. The adsorption amount of rhodamine B for two hours showed that the q_e_ of TiO_2_@HNT/Alcap was 0.1704, which was about 1.7 times higher than that of the bare alginate capsules (q_e_ 0.1026). This result indicated that TiO_2_@HNT/Alcap was 0.1704, which was about 1.7 times higher than that of the bare alginate capsules (q_e_ 0.1026). This result indicated that the TiO_2_@HNTs inside the alginate gel could expand the surface area to bind the dye molecules, enhancing the adsorption ability.

### 3.4. Photocatalytic Degradation of Rhodamine B Using TiO_2_@HNT/Alcaps

After confirmation of enhanced adsorption of the catalysts, we monitored the photocatalytic degradation of the rhodamine B under UV conditions (Figure 6a). The solution of rhodamine B and the catalysts became transparent, indicating the photolysis of dyes after 2 h. The TiO_2_@HNT/Alcaps showed a 97.65% decomposition of rhodamine B for 120 min, while the alginate gel, HNT/Alcap, and TiO_2_/Alcap showed only a maximum of 50% degradation. Since the HNT/Alcaps can adsorb the dye molecules but are unable to decompose the dye due to the lack of photocatalysts, only adsorption occurred, without photolysis. Although the TiO_2_/Alcaps showed higher degradation efficiency than the HNT/Alcaps, they showed only 50% photocatalytic efficiency compared to the TiO_2_@HNT/Alcaps. From this result, we assumed that the alginate gels provide a supporting site facilitating the prevented aggregation of TiO_2_ nanoparticles but that they cannot offer the effect of enhanced catalytic activity, unlike the HNTs. In the case of the TiO_2_@HNT/Alcaps, the alginate gel provides a millimeter-sized structure for easy separation; the HNTs offer enhanced adsorption efficiency, inducing the improved photocatalytic activity, and the TiO_2_ nanoparticles as photocatalysts take charge of photolysis. This hierarchical structure based on their function allowed the TiO_2_@HNT/Alcaps to boost photocatalytic activity. 

From kinetic analysis, we could confirm that the catalysis followed the pseudo-first-order kinetics. (Figure 6b and Table 2) [43]. The rate constant *k_1_* of TiO_2_@HNT/Alcap was 10 times and 6 times higher than that of HNT/Alcap and TiO_2_/Alcap, respectively, showing improved degradation efficiency. 

### 3.5. Catalyst Recycling for Photocatalytic Degradation 

In the catalyst recycling experiment, our catalysts were separated using a strainer after photodegradation under UV conditions. The catalysts’ separation time from the reaction mixture was only one minute using a strainer (mesh size: 1 mm). The separation time can be shortened by increasing the water flow rate and broadening the mesh size of the strainer. Although the TiO_2_@HNT/Alcaps had lower photocatalytic activity (*k*_1_: 0.0321 min^−1^) compared to the nanocatalysts (TiO_2_@HNTs, *k*_1_: 0.1206 min^−1^) [19], our catalysts possess economic benefits for the treatment of a large amount of wastewater, considering that the centrifugation of a large amount of wastewater would require huge equipment and energy. Without the centrifugal separation process, the TiO_2_@HNT/Alcaps could maintain their photocatalytic activity at under 10% loss in five cycle reuses, showing the potential of reusable catalysts for industrial use (Figure 7 and Appendix A). 

## 4. Conclusions

In this work, we encapsulated the TiO_2_@HNTs catalysts into alginate gel for fast separation. The hybrid capsule was analyzed by FE-SEM to confirm the alginate surface morphology and the encapsulation of TiO_2_@HNTs inside the gels. The thermal decomposition properties and the composition ratio of the capsule were confirmed by TGA analysis. The adsorption of rhodamine B using TiO_2_/HNT@Alcap increased by 1.7 times over that of the alginate gel, indicating the expansion of the surface area to which the dye can bind. The photocatalytic activity of TiO_2_/HNT@Alcap under UV conditions was improved by six times compared to that of TiO_2_/Alcaps. The retrieval strategy was demonstrated by easy and fast separation of our catalyst in five recycling tests with a 1 min separation time using a strainer. The stability of the catalyst was demonstrated through its high removal efficiency, with under 10% loss of function over five times of recycled use. This designed catalyst with retrieval capacity and catalytic efficiency can be applied to purify an aquatic environment contaminated with organic pollutants for the textile industry.

## Figures and Tables

**Figure 1 nanomaterials-12-02361-f001:**
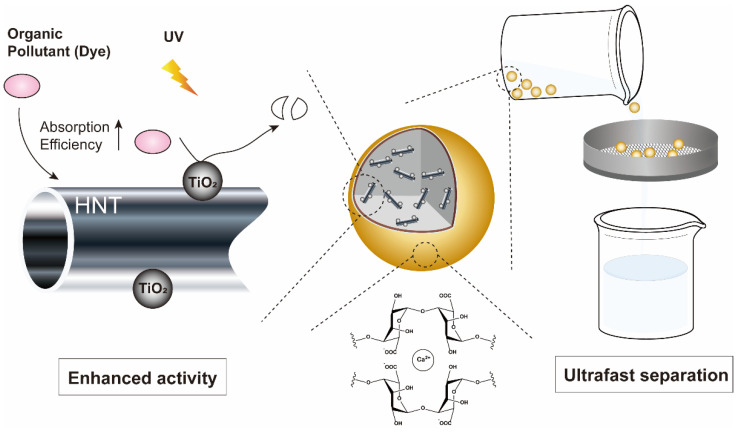
Design of TiO_2_@HNTs/Alcaps and graphical scheme of their easy separation. The combination of TiO_2_ nanoparticles, HNTs, and encapsulation in alginate gel improves the photocatalytic activity through enhanced adsorption efficiency and high reusability through the easy separation using a strainer.

**Figure 2 nanomaterials-12-02361-f002:**
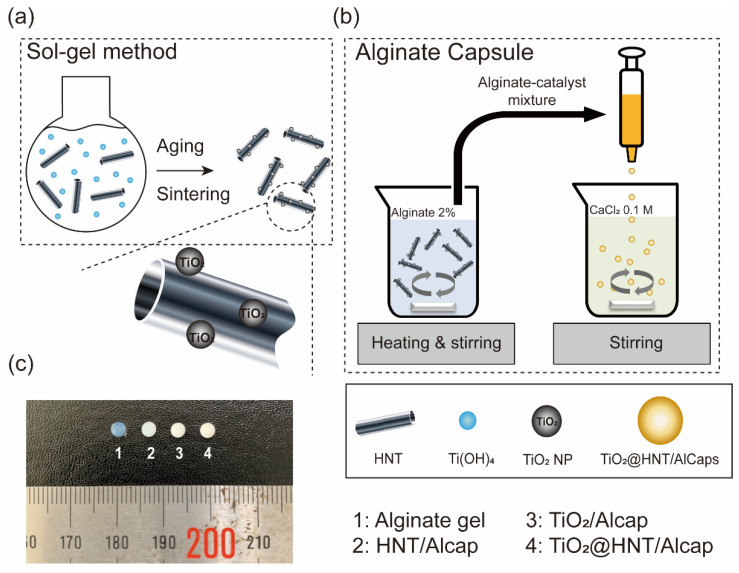
Synthesis processes of TiO_2_@HNTs nanocomposites and TiO_2_@HNTs/Alcaps: (**a**) TiO_2_ nanoparticles on the HNT surface were synthesized using the sol-gel method. (**b**) The alginate capsules were prepared by dropping the mixture of alginate-TiO_2_@HNTs to CaCl_2_ solution. The calcium ions binding to alginate polymer chain form the cross-linked gel. (**c**) Photos of 1: alginate gel; 2: HNT/Alcap; 3: TiO_2_/Alcap; and 4: TiO_2_@HNT/Alcap.

**Figure 3 nanomaterials-12-02361-f003:**
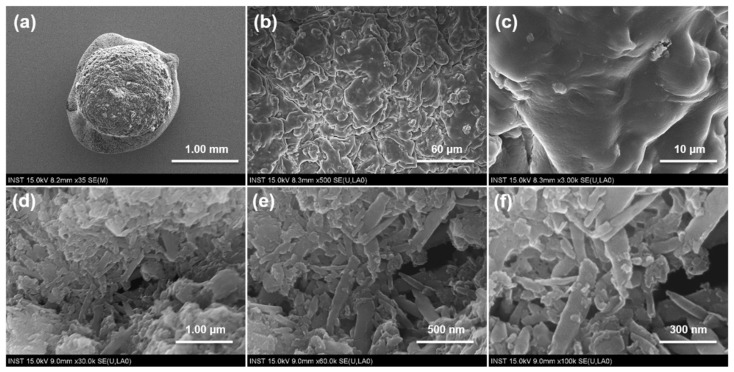
Scanning electron microscopy images of TiO_2_@HNT/Alcaps. Surficial morphology (**a**–**c**) and the interior structure of TiO_2_@HNT/Alcaps (**d**–**f**) were visualized.

**Figure 4 nanomaterials-12-02361-f004:**
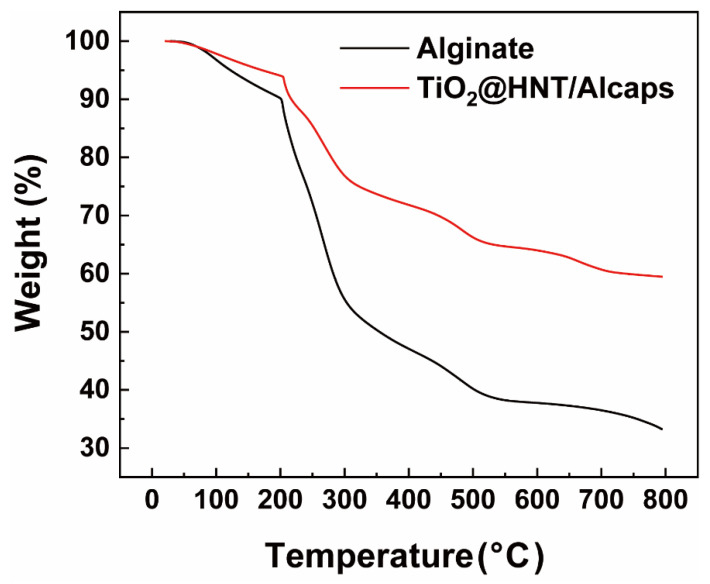
TGA curves of alginate gel and TiO_2_@HNT/Alcaps.

**Figure 5 nanomaterials-12-02361-f005:**
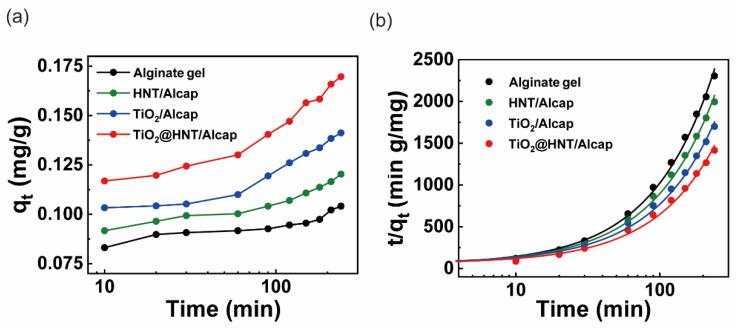
(**a**) Adsorption of rhodamine B onto alginate gel, HNT/Alcap, TiO_2_/Alcap, and TiO_2_@HNT/Alcaps. (**b**) Pseudo-second-order kinetics for rhodamine B adsorption on the catalysts.

**Figure 6 nanomaterials-12-02361-f006:**
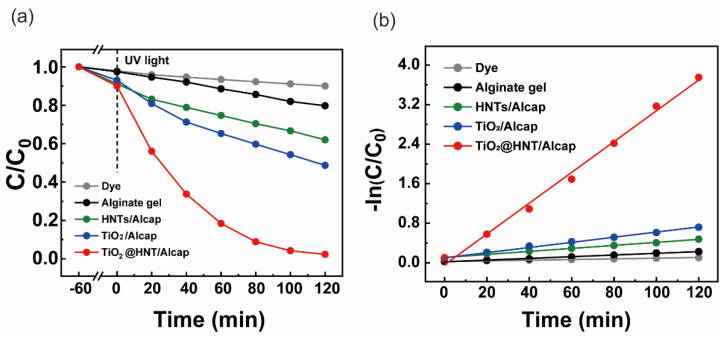
(**a**) Photocatalytic degradation of rhodamine B with various alginate/catalyst hybrid capsules under UV light. (**b**) Pseudo-first-order kinetic curves of photocatalytic degradation of rhodamine B, depending on catalysts.

**Figure 7 nanomaterials-12-02361-f007:**
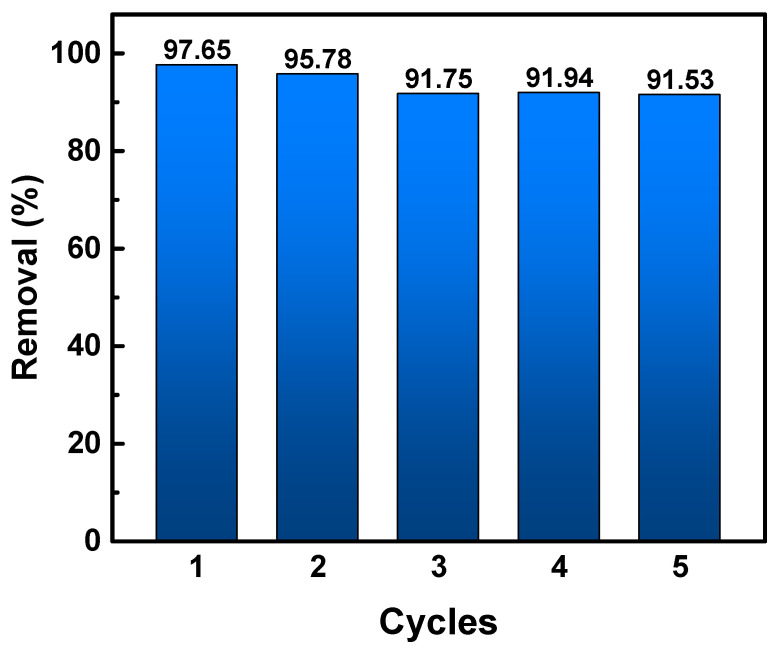
The total removal % of dye in several successive cycles of reaction.

**Table 1 nanomaterials-12-02361-t001:** Pseudo-second-order kinetic model parameters of rhodamine B adsorption by catalyst composite capsules.

Sample	q_e_ (mg/g)	*k*_2_(g/mg min)	R^2^
Adsorption			
Alginate	0.1026	1.9690	0.9967
HNT/Alcap	0.1191	1.2501	0.9961
TiO_2_/Alcap	0.1420	0.7619	0.9948
TiO_2_@HNT/Alcap	0.1704	0.5635	0.9932

**Table 2 nanomaterials-12-02361-t002:** Pseudo-first-order kinetic parameters of photocatalytic degradation of rhodamine B by catalyst composite capsules.

Sample	Dye removal %	*k*_1_ (min^−1^)	R^2^
UV irradiation			
Dye	9.980	0.0007	0.9918
Alginate gel	20.30	0.0017	0.9963
HNT/Alcap	38.01	0.0031	0.9943
TiO_2_/Alcap	51.32	0.0052	0.9933
TiO_2_@HNT/Alcap	97.65	0.0312	0.9939

## Data Availability

The data presented in this study are available in this article and the Appendix A.

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
