# Peer review of "Strainer-Separable TiO2 on Halloysite Nanocomposite-Embedded Alginate Capsules with Enhanced Photocatalytic Activity for Degradation of Organic Dyes"

_nanomaterials, 2022, doi:10.3390/nano12142361_

Round 1
Reviewer 1 Report
In the manuscript “Strainer-separable TiO2 on Halloysite Nanocomposites- Embedded Alginate Capsules with Enhanced Photocatalytic Activity for Degradation of Organic Dyes” the authors prepared an alginate capsule including TiO2 nanoparticles on HNTs (TiO2@HNTs/Alcaps) to achieve both improved photocatalytic activity for degradation of organic dyes and ultra-fast catalyst separation.
Overall, the analyses of the data are adequate for the above mentioned issue.
The conclusions are relevant and supported by the results, being of interest for the readership of Nanomaterials journal.
Accordingly, I consider this appropriate for publication, taking into account the next comments and suggestions.
1. “The TEM images of the TiO2@HNT clearly show the immobilization of TiO2 nanoparticles on the HNTs and the crystal structure of nanoparticles” – supplementary confirmation by the X-ray diffraction analysis it would be necessary.
2. “Considering that the synthesis ratio of alginate and TiO2@HNT is 1:1(w/w), the authors estimate that our TiO2@HNT/Alcaps consisted of 13% of HNTs, 13% of TiO2, and 74% of alginate gel” - The authors should calculate the composition using XPS or ICP analysis.
Author Response
See the attched author reply.

Reviewer 2 Report
Dear Editor: I would like to express my deep thanks for inviting me to review the manuscript ID: nanomaterials-1807670-peer-review-v1
Title: SStrainer-separable TiO2 on Halloysite Nanocomposites-Embedded Alginate Capsules with Enhanced Photocatalytic Activity for Degradation of Organic Dyes
Authors: Jewon Lee, Sicheon Seong, Soyeong Jin, Jaeyong Kim, Youngdo Jeong, and Jaegeun Noh
Comments:
Abstract:
Rewrite the abstract according to the results.
Introduction:
Need to rewrite introduction part and emphasize the importance of Strainer-separable TiO2 on Halloysite Nanocomposites.
“First, a TiO2 on HNTs (TiO2@HNTs) composite was synthesized 74 by a simple sol-gel method using titanium(IV) isopropoxide (TTIP) precursor”.
Replaced by
“There are different chemical methods uses for the synthesis of uniform micro- and nano-particles and their composites for example micro-emulsion method [1], co-precipitation method [2], sol-gel method [3], sol-vothermal/hydrothermal method with different surfactant and caping agents [4]. In this work, a TiO2 on HNTs (TiO2@HNTs) composite was synthesized by a simple sol-gel method using titanium(IV) isopropoxide (TTIP) precursor”.
1. J. Nan, C. Huang, L. Tian, C. Shen, Effects of micro-emulsion method on microwave dielectric properties of 0.9Al2O3-0.1TiO2 ceramics Mater. Lett., 249 (2019) 132-135.
2. B. Shivaraj, M.C. Prabhakara, H.S. B. Naik, E. I. Naik, R. Viswanath, M. Shashank, B.E. K. Swamy, Optical, bio-sensing, and antibacterial studies on Ni-doped ZnO nanorods, fabricated by chemical co-precipitation method, inorganic Chemistry Communications 134 (2021) 109049
3. B.T. Lee, J.K. Han, A.K. Gain, K.H. Lee, F. Saito, “TEM microstructure characterization of nano TiO2 coated on nano ZrO2 powders and their photocatalytic activity” Materials Letters 60 (17-18), (2006) 2101-2104
4. J. Hongquan, L. Yanduo, L. Jingshen, W. Haiyan, Synergetic effects of lanthanum, nitrogen and phosphorus tri-doping on visible-light photoactivity of TiO2 fabricated by microwave-hydrothermal process, Journal of Rare Earths, 34(6), (2016) 604-613.
Please explain novelty and objectives of this work
Materials and Methods
Briefly explain the synthesis method
Explain in detail the characterization section
Results and discussion:
Explain in detail the SEM images in Figure 3 (d-f) and emphasize on the formation mechanism of rod-shape structure. Please provide EDS analysis data in In Figure 3.
Figure 4 does not provide new information according to your references 35-37.
It is very difficult to understand X-axis scale starting point -60minutes in Figure 6(a). Further there is no detail explanation why TiO2@HNTs dramatically enhanced adsorption efficiency.
Please added al data in Figure 7.
Conclusion part:
Please rewrite the conclusion part in bullet points.
RECOMMENDATION
After reviewing the enclosed manuscript for “Nanomaterials”, the present manuscript contains some kinds of scientific analysis but it is mandatory required to modify according to the preceding remarks. So, the manuscript can be publication after major revision.
Author Response
See the attched author reply.

Round 2
Reviewer 2 Report
Authors addressed all the comments in the revised manuscript.